# Benchmarking In-the-Wild Multimodal Plant Disease Recognition and A Versatile Baseline

## ABSTRACT

Existing plant disease classification models have achieved remarkable performance in recognizing in-laboratory diseased images. However, their performance often significantly degrades in classifying in-the-wild images. Furthermore, we observed that in-the-wild plant images may exhibit similar appearances across various diseases (*i.e.*, small inter-class discrepancy) while the same diseases may look quite different (*i.e.*, large intra-class variance). Motivated by this observation, we propose an in-the-wild multimodal plant disease recognition dataset that contains the largest number of disease classes but also text-based descriptions for each disease. Particularly, the newly provided text descriptions are introduced to provide rich information in textual modality and facilitate in-the-wild disease classification with small inter-class discrepancy and large intra-class variance issues. Therefore, our proposed dataset can be regarded as an ideal testbed for evaluating disease recognition methods in the real world. In addition, we further present a strong yet versatile baseline that models text descriptions and visual data through multiple prototypes for a given class. By fusing the contributions of multimodal prototypes in classification, our baseline can effectively address the small inter-class discrepancy and large intra-class variance issues. Remarkably, our baseline model can not only classify diseases but also recognize diseases in few-shot or training-free scenarios. Extensive benchmarking results demonstrate that our proposed in-the-wild multimodal dataset sets many new challenges to the plant disease recognition task and there is a large space to improve for future works.

## CCS CONCEPTS

• **Computing methodologies** → **Object recognition**.

## KEYWORDS

Plant disease, Vision language models

## 1 INTRODUCTION

Plants often face threats from a wide range of diseases caused by bacteria, pests, and viruses. The Food and Agriculture Organization of the United Nations estimates annual losses of approximately 220 billion dollars due to plant diseases [1]. Accurate recognition of plant diseases is essential to mitigate damage and prevent the spread of these diseases.

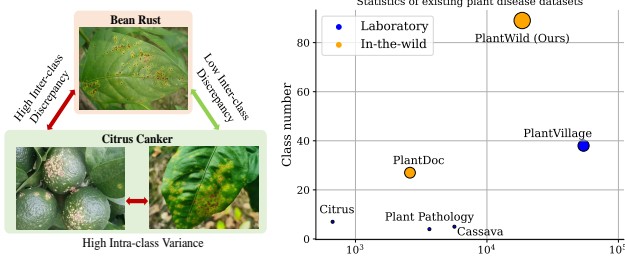

**Figure 1: Left: Illustration of intra-class variances and inter-class discrepancies among plant disease images. Right: statistics of existing plant disease datasets. The marker size corresponds to the number of plant diseases. Our proposed PlantWild dataset not only encompasses the largest number of disease classes but also includes the highest volume of in-the-wild images.**

Recent deep learning-based plant disease recognition methods have achieved promising performance on in-laboratory images [11, 19, 27]. However, they would suffer drastic performance degradation when plant images are captured in the wild [29] due to the large domain gap caused by complex backgrounds, viewpoints, and lighting conditions. The existing in-the-wild plant disease recognition dataset PlantDoc [29] only contains 2,598 images with 27 disease types, and thus it might not meet the practical needs in the real world. Therefore, it is necessary to collect a large-scale in-the-wild plant disease dataset with more common disease types.

As illustrated in Figure 1, we observed that in-the-wild plant images sometimes exhibit very similar appearances across different diseases, known as small inter-class discrepancies, while the same disease shows very distinct appearances, known as large intra-class variances. Therefore, we speculate that a deep classification model would struggle to accurately distinguish different plant diseases solely based on images. Unlike the previous in-the-wild disease dataset [29] that only contains disease labels as ground truth, we supplement descriptive textual prompts for each disease. The prompts can provide more discriminative information to assist plant disease recognition.

In this work, we curate a large-scale multimodal in-the-wild plant disease dataset PlantWild that contains not only diseased and healthy plant images but also multiple text descriptions for each class, as shown in Figure 2. Specifically, the image data are crowd-sourced from diverse internet sources and the descriptions of each class are obtained from Wikipedia and GPT-3.5 [4]. To ensure the quality of images and the correctness of the corresponding labels, we invite five annotators to filter low-quality or irrelevant images of our crowd-sourced data and then annotate image labels (*i.e.*, disease classes). The labels and text descriptions of collected images are checked at least by two annotators and verified by an expert. In total, we collect 18,542 plant images that are captured in various

viewpoints, lighting conditions and backgrounds, and contain 89 disease types. Our PlantWild dataset is the largest in-the-wild plant disease dataset in terms of image numbers and disease classes.

To better understand the new challenges posed by PlantWild, we provide a strong yet versatile baseline method. Our proposed baseline is introduced to fully exploit the multimodal information provided by PlantWild and support various disease recognition scenarios, such as training-free and few-shot learning. As observed, the intra-class variances could be very large. Thus, we opt to model each class with multiple prototypes instead of only one prototype. Specifically, we leverage CLIP [22] to extract visual features from training images and then group the features within each class to generate multiple prototypes. As multiple visual prototypes can cover the majority of visual characteristics of a class, the intra-class variance problem can be significantly mitigated.

Furthermore, considering inter-class discrepancies might be small in visual data, directly learning from plant images does not lead to discriminative decision boundaries. Therefore, we resort to the textual modality to aid our disease recognition model in learning discriminative boundaries. To be specific, we also extract textual features from the text descriptions of each class via the CLIP textual encoder and then employ the textural features of each class as the textual prototype. In this fashion, visually hard-to-distinguishing samples can still be semantically separated in the textual feature space. This further motivates us to take both visual and textual features into account for final classification. Since text descriptions are not available in the testing phase, we feed visual features into both the visual and textural prototypes and then fine-tune the prototypes with the classification losses. Thanks to the multimodal prototypes, our baseline model can be applied to conventional classification tasks as well as few-shot and training-free ones.

We conduct extensive experiments on our proposed PlantWild dataset and benchmark state-of-the-art methods in the conventional classification task as well as few-shot and training-free situations. Additionally, we also evaluate another two plant disease datasets, *i.e.*, PlantVillage and PlantDoc. Moreover, although our versatile baseline model outperforms the state-of-the-art in various tasks, PlantWild also imposes a plethora of challenges to current plant disease recognition, such as accurate identification of disease areas from complex backgrounds, large intra-class appearance variances and small inter-class discrepancies. These challenges further manifest the practicality and necessity of our PlantWild dataset.

## 2 RELATED WORK

### 2.1 Plant Disease Classification

Plant disease classification [14, 29] is critical to prevent the spread of disease among plants. PlantVillage [11] is the most widely used plant disease dataset. It contains 54,309 images of 38 classes. All the images in PlantVillage are captured in laboratory environments, thus lacking complex backgrounds in the wild conditions. In contrast, PlantDoc [29] consists of 2,598 wild images of 27 categories. As its images are collected from Internet sources, it usually has diverse backgrounds and accounts to the real-world complexity. Built on those datasets, Ramesh et al. [26] propose a bi-linear convolution neural network that consists of two pathways to produce the plant disease representations. Wang et al. [35] further present a dual-stream hierarchical bilinear pooling model, which leverages the interaction between the last few layers from the two pathways. Wang et al. [34] introduce a trilinear convolutional neural network model by utilizing three CNNs as its base network. Borhani et al. [3] propose a vision transformer-based method that combines both attention blocks and CNN blocks to improve the recognition speed. More recently, Joseph et al. [14] investigate how the diseases affect eleven plants and how the diseases can be identified from plant leaf images using CNN-based models.

### 2.2 Vision-Language Modeling

Large pre-trained Vision-Language Models (VLMs) [2, 5, 7, 12, 17, 22] have been developed to bridge the gap between vision and language modalities. CLIP [22] is the most widely-used VLM and contains two encoders to project images and texts into the same joint embedding space. Furthermore, CLIP illustrates a remarkable zero-shot ability for classification tasks.

Some CLIP-based follow-up approaches introduce extra learnable parameters to improve performance on downstream datasets. Specifically, Coop [40], CoCoop [39], LiFT [16], TPT [28] and MaPLe [15] optimize learnable prompts rather than using hand-crafted prompts. CLIP-Adapter [8] introduces feature adapters to fine-tune the features extracted by CLIP encoders. These methods can greatly improve classification performance on downstream tasks by optimizing the learnable parameters with a few training data, while the parameters of the original CLIP model are kept frozen. Some cache-based methods like Tip-Adapter [38], CaFo [37] and Sus-X [32] have been proposed. They usually store features extracted from few-shot training samples and make predictions based on the affinities of the stored features and test features. These methods achieve classification in a training-free manner, while their performance can be further boosted if training is available. However, these methods are proposed to address generic zero/few-shot or conventional fully supervised problems, and do not fully take the characteristics of plant diseases (*i.e.*, high inter-class similarity and high intra-class diversity) into account, and thus direct employment of these methods to plant disease recognition does not lead to satisfactory results.

### 2.3 Prompt Design with Large Language Models

Large language models (LLMs) [4, 23–25, 31] have made significant achievements in processing natural languages and demonstrated remarkable capacity in text generation. This superior capability has been employed by some works [37, 41] via prompts for downstream tasks. For instance, CLIP utilizes manually crafted prompts as input for its textual encoder. These prompts are usually structured as "a photo of a [CLASS]", where [CLASS] denotes the name of a category. Previous research [13, 21, 36] has demonstrated that incorporating prior knowledge into prompts can enhance the performance of zero-shot learning. By fully exploiting linguistic knowledge and aligning visual features to textual semantics, we speculate textual semantics would help to solve some visually hard-to-distinguished cases. Motivated by this, we employ GPT-3.5 [4] to produce text descriptions of each disease with diverse prompts in our PlantWild.

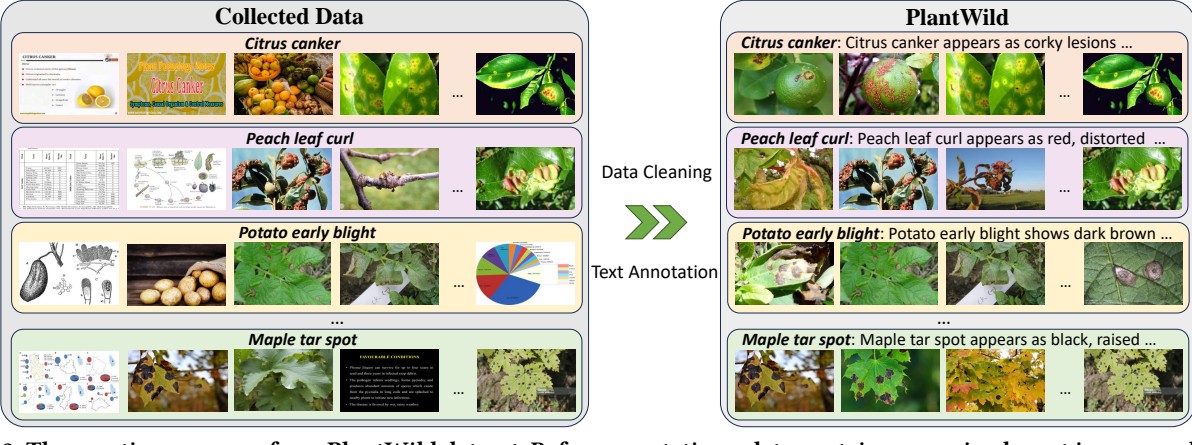

**Figure 2: The curation process of our PlantWild dataset. Before annotations, data contains many irrelevant images and is very noisy. After annotations, PlantWild consists of in-the-wild disease-relevant images and text descriptions for each class.**

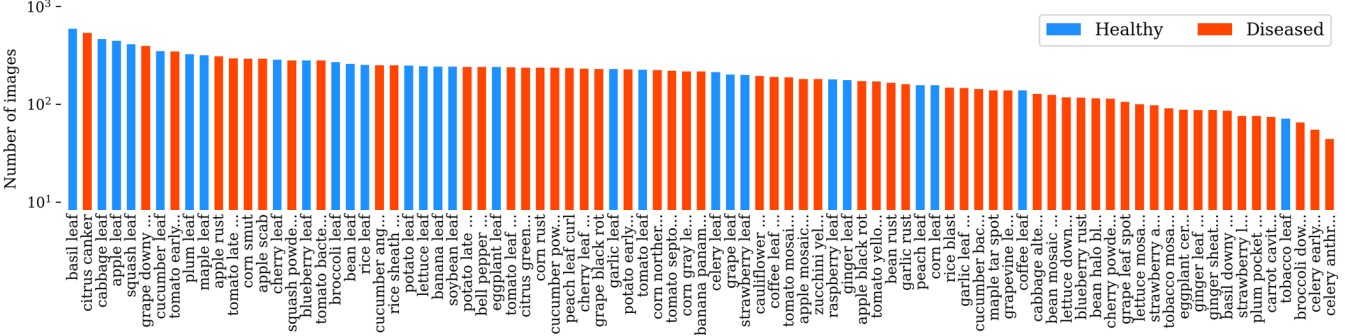

**Figure 3: Illustration of statistics of our PlantWild dataset. It contains 56 diseased and 33 healthy classes. The number of images within a class ranges from 589 to 44 images.**

## 3 METHODOLOGY

### 3.1 Proposed PlantWild Dataset

*3.1.1 Data Collection and Filtering.* Existing in-the-wild plant disease recognition dataset falls short in scale. Motivated by this fact, we curate a large-scale in-the-wild plant disease recognition dataset in the wild, covering a wide range of categories. Based on previous plant disease datasets [11, 29] and other common plant species, we determine 89 classes, including 33 healthy plant classes and 56 diseased classes. Following the convention [29], we collect our data by downloading images from Google Images[1] and Ecosia[2]. In addition, we also download images from Baidu Images, the largest image search engine in Chinese. As China has vast farms, covering a wide range of areas and crops, sources from Chinese websites can greatly expand our dataset. We employ the common names of each class in English and Chinese as keywords for querying images. From these sources, we have collected more than 50,000 images in total. As shown in the left side of Figure 2, there are many irrelevant images and thus cleaning the data is necessary.

Our annotators filter the data mainly based on the image exemplars of plant diseases from UMN[3], *i.e.*, removing erroneous images

from the respective folders of disease classes. To ensure annotation accuracy, each image is cross-validated by at least two experts. If the two annotators cannot reach a consensus on a particular image, we invite another expert to review and correct the annotations. The resulting dataset consists of 18,542 images across 89 classes. The class with the most images includes 589 images and the class with the fewest images includes 44 images. The statistics of the dataset is shown in Figure 3. To the best of our knowledge, our PlantWild dataset is the largest in-the-wild plant disease recognition dataset and also contains the most classes.

*3.1.2 Textual Prompt Generation.* In the field of plant diseases, certain diseases may present similar overall appearances but differ in specific aspects. For instance, both potato early blight and late blight result in brown spots on leaves. However, early blight causes roughly round spots and there are concentric circular patterns within the spots, while late blight spots are generally irregular and look like water-soaked. To capture such fine-grained features, we provide rich text descriptions (*i.e.*, textual prompts) for each class in PlantWild. Some works [21, 37] indicate that descriptive prompts with prior knowledge can help visual representations to focus on fine-grained image details, thus the classification performance is benefited. Motivated by this, we leverage the large-scale language model GPT-3.5 [4] to generate descriptive prompts. The ample

[1]https://www.google.com/
[2]https://www.ecosia.org/?c=en
[3]https://extension.umn.edu/solve-problem/plant-diseases

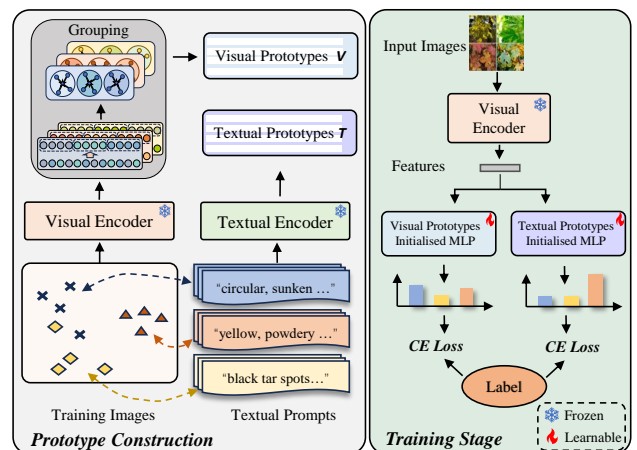

**Figure 4: Overall architecture of our baseline. CLIP encoders extract features from images and text for each category and then multiple prototypes are constructed by grouping visual features. Given a test image, both the visual and textual prototypes can be used for classification.**

semantic information of these prompts can help identify the subtle differences between different plant diseases. Considering the visual diversity within the same category in our dataset, relying on a single prompt for each class might not sufficiently cover all representative features. Therefore, we generate multiple prompts for each category, where different text prompts focus on describing diverse visual features of a class. We will release both the images and the prompts of our PlantWild datasets.

*3.1.3 Practicality and Necessity of PlantWild.* Many existing datasets only focus on diseases that affect specific crops, such as citrus [27], apple [30], and cassava [19]. The most widely-used dataset, PlantVillage [11], includes 38 classes across 14 plant species, thus it provides generalizability across different species. However, PlantVillage consists solely of laboratory images. Those images are taken under controlled environments at experimental research stations, devoid of real-world complex background information. In response, PlantDoc [29] is introduced. PlantDoc contains wild images sourced from the internet and is more suitable for training models in real-world plant disease recognition. Nonetheless, PlantDoc comprises only about 2,600 images, significantly smaller than PlantVillage in scale. Furthermore, the plant species and diseases in PlantDoc are also limited as it only includes 27 classes across 13 plant species. In contrast, our PlantWild dataset is the largest in-the-wild image dataset and it will be public available. It includes a significantly broader range of categories compared to existing plant disease datasets. Additionally, PlantWild is enriched with textual annotations for all 89 categories, providing a multi-modal annotation framework for potential performance improvements.

## 3.2 Multimodal Versatile Plant Disease Recognition Baseline

We propose a multimodal versatile baseline, named MVPDR, for effective plant disease recognition. The overall framework of our baseline is illustrated in Figure 4 and is also elaborated in Algorithm

---

**Algorithm 1** Prototype construction and training procedure of MVPDR

**Input:** Training images $I$, descriptive prompts $P$.
**Output:** Visual prototypes $V$, textual prototypes $T$
1: $\{F^v, T\} \leftarrow$ Extract features from $\{I, P\}$ using CLIP
2: **Phase 1**: Prototype construction:
3: **for** class m = 1, ..., $M$ **do**
4:     Initialize $k$ cluster centroids randomly: $C_{m1}, C_{m2}, ..., C_{mk}$
5:     Assign each visual vector $f_m^{v(i)}$ to the nearest centroid and update each centroid with the mean of the features:
6:     $C_{mi} \leftarrow \frac{1}{N^i} \sum_{i=1}^{N^i} f_m^{v(i)}$, $C_{mi}$ is assigned with $N^i$ samples
7:     Repeat steps 5 and 6 until convergence
8:     $V \leftarrow$ Concatenate all the feature vectors of centroids $C_{mi}$
9: **end for**
10: **Phase 2**: Training the obtained prototypes:
11: $V \leftarrow$ Train visual prototypes by optimizing Eq.4
12: $T \leftarrow$ Train textual prototypes by optimizing Eq.5 and Eq. 6
13: **return** $V, T$

---

1. Specifically, MVPDR utilizes visual data and text descriptions to construct prototypes for classification. Visual prototypes are obtained by clustering image representations per class, and thus they can store information of visual characteristics. Textual prototypes are based on descriptive text prompts, which provide semantic information to better identify the symptoms of different plant diseases. MVPDR combines the contributions of the multimodal prototypes to enhance accuracy.

*3.2.1 Prototype Construction.* We respectively construct the visual and textual prototypes with the visual and the textual encoder of CLIP [22]. Given a dataset with $M$ classes, all images $I = \{I_i\}_{i=1}^{N_{train}}$ in the training set are fed into the CLIP's visual encoder to obtain the image features $F^v \in \mathbb{R}^{N_{train} \times D}$, where $N_{train}$ is the number of all training samples and $D$ denotes the dimension of each feature vector. We iteratively apply the grouping technique K-means [18] for each class to get $K$ clusters, thus the image features with similar representations are clustered together. As containing representative visual characteristics from training samples, every centroid of these clusters is employed as a visual prototype. The prototypes are assigned to the corresponding class where they are obtained from. Therefore, we have $M \times K$ visual prototypes in total, and they are stored as prototypes $V \in \mathbb{R}^{M \times K \times D}$. On the other hand, we leverage the textual encoder of CLIP to extract textual features from the descriptive prompts $P \in \mathbb{R}^{M \times J}$, where $J$ represents the number of prompts for each class. With multiple prompts for each class, we directly employ the textual features to construct the textual prototypes $T \in \mathbb{R}^{M \times J \times D}$.

*3.2.2 Multimodal Prototype Learning.* The constructed prototypes can serve as initialized weights of a classifier for recognition. To classify an image $I$, MVPDR first extracts it into a feature vector $x \in \mathbb{R}^D$ using the CLIP's visual encoder. Afterward, the cosine similarity matrices between $x$ and all the prototypes can be acquired as $\cos(x, V_i^k)$ and $\cos(x, T_i^j)$, where $V_i^k$ and $T_i^j$ represent the $k$-th visual and $j$-th textual prototype of the $i$-th category. $V_i^k, T_i^j$ and $x$ are L2 normalized, thus the cosine similarity values vary from 0 to 1.

Typically, the prediction is assigned to the category of the prototype that is most similar to the input feature. As each class has multiple visual and textual prototypes, we take several integrating strategies for visual and textual prototypes to obtain class-level prediction. To produce the classification logits via the visual prototypes, we sum the cosine similarity of the prototypes in each class and calculate the probability of $i$-th class using Softmax:

$$p_{v_i} = \frac{\exp\left(\sum_{k=1}^{K} \cos(x, V_i^k)/\tau\right)}{\sum_{m=1}^{M} \exp\left(\sum_{k=1}^{K} \cos(x, V_m^k)/\tau\right)}, \qquad (1)$$

where $\tau$ denotes the temperature factor. In terms of textual prototypes, we separately obtain two logits by conducting a maximum operation and average operation for each class on the similarity matrix of textual prototypes. The maximized probability logits are computed as:

$$p_{\text{tmax}_i} = \frac{\exp\left(\cos(x, T_i^{\tilde{j}})/\tau\right)}{\sum_{m=1}^{M} \exp\left(\cos(x, T_m^{\tilde{j}})/\tau\right)}, \qquad (2)$$

where $\tilde{j} = \arg\max_j \cos(x, T_i^j)$, representing the index of the prototype with the highest similarity score. $p_{\text{tmax}_i}$ only focus on the prototype that is most similar to the input feature vector but may overlook useful information contained in other prototypes. We also obtain the averaged logits to fully utilize all the text features. The averaged probability of $i$-th class can be written as:

$$p_{\text{tavg}_i} = \frac{\exp\left(\sum_{j=1}^{J} \cos(x, T_i^j)/(\tau * J)\right)}{\sum_{m=1}^{M} \exp\left(\sum_{j=1}^{J} \cos(x, T_m^j)/(\tau * J)\right)}. \qquad (3)$$

Prototypes $V$ and $T$ can be regarded as weights of classifiers. Extracted from training images and descriptive textual prompts, they preserve the prior knowledge from the pre-trained CLIP and thus can benefit the classification performance. To further improve the accuracy, we set the prototypes as learnable parameters. We leverage cross-entropy losses for obtained logits to train the prototypes while keeping the parameters of the CLIP backbone frozen. The losses corresponding to $p_{v_i}$, $p_{\text{tmax}_i}$ and $p_{\text{tavg}_i}$ are as follows:

$$L_v = -\frac{1}{N_{train}} \sum_{n=1}^{N_{train}} \sum_{i=1}^{M} y_i^n \log p_{v_i}^n, \qquad (4)$$

$$L_{tmax} = -\frac{1}{N_{train}} \sum_{n=1}^{N_{train}} \sum_{i=1}^{M} y_i^n \log p_{\text{tmax}_i}^n, \qquad (5)$$

$$L_{tavg} = -\frac{1}{N_{train}} \sum_{n=1}^{N_{train}} \sum_{i=1}^{M} y_i^n \log p_{\text{tavg}_i}^n, \qquad (6)$$

where $N_{train}$ is the number of training samples, $y_i = 1$ if $i$ equals to the ground truth label, otherwise $y_i = 0$. The overall loss function is given by:

$$L_{overall} = L_v + \lambda_1 \cdot L_{tmax} + \lambda_2 \cdot L_{tavg}, \qquad (7)$$

where $\lambda_1$ and $\lambda_2$ are the weights for the loss functions.

3.2.3 *Inference.* With the prototypes trained on training images, we can conduct classification according to the probability logits obtained in Eq. 1, 2 and 3. To improve the classification performance, we integrate each output probability by adjusting three hyper-parameters. The overall logits are ensembled as:

$$p_i = \alpha \cdot p_{v_i} + \beta \cdot p_{\text{tmax}_i} + \gamma \cdot p_{\text{tavg}_i}, \qquad (8)$$

where $p_i$ denotes the final prediction probability of the $i$-th category. $\alpha$, $\beta$ and $\gamma$ are hyper-parameters to balance the weights of each probability. Among $M$ classes, the class with the highest probability value is then selected as the final prediction.

## 4 EXPERIMENTS

### 4.1 Experiment Setups

**Datasets and evaluation metrics** We conduct experiments on our curated PlantWild dataset, as well as other two existing plant disease datasets PlantDoc [29] and PlantVillage [11]. Among the three datasets, PlantVillage is curated under laboratory environments while PlantWild and PlantDoc contain in-the-wild images. Our datasets are split into three subsets: training set, validation set, and test set, with ratios of 70%, 10%, and 20% respectively. For PlantDoc and PlantVillage, we follow the given standard for the data split. Following the conventions of classification tasks, we use Accuracy (marked by Acc) to evaluate the performance of our proposed method and existing methods. In addition, we also apply Macro precision (denoted by M-P) and Macro recall (denoted by M-R) to evaluate the methods' performance by considering false positives and false negatives in the dataset. Macro precision assesses the ability to identify true positives across all classes, while Macro recall assesses the ability to identify all relevant instances. Marco F1-score (*i.e.*, the average F1 scores across all the classes, marked by M-F1) is also reported to provide a balanced assessment by considering both precision and recall.

**Implementation** The core of our baseline involves extracting features from descriptive texts and training images using CLIP [22] and then constructing prototypes in both modalities. For CLIP's backbones, we employ ResNet101 [10] as the image encoder and a transformer for the textual encoder to generate prototypes. We conduct few-shot, and fully-supervised experiments to evaluate the recognition ability of a model in different scenarios. Under few-shot scenarios, we randomly select 16 samples for each class for visual prototype construction and training, while all training images are available in the fully supervised experiment setting. During the training process, we set the visual and textual prototypes to be learnable while all the parameters of CLIP backbones are kept frozen. We set the batch size to 64 and train our model for 30 epochs. We choose the AdamW optimizer with a cosine scheduler for both visual and textual prototypes. The initial learning rate is set to 0.003. In addition, we also explore the training-free classification capability of our method. When training is prohibited, we leverage the initially constructed prototypes and descriptive texts for classification.

**Competing baselines** As our baseline method is built on CLIP, we also include CLIP-based methods (such as CoOp [40], CLIP-Adapter [8], Tip-Adapter [38]). Furthermore, state-of-the-art plant disease classification methods (T-CNN [34], DHBP [35]) are also employed to evaluate our PlantWild dataset. Note that, state-of-the-art plant disease classification methods are in general trained in a fully supervised manner. Thus, their results are mainly reported under fully supervised experiment settings. For the CLIP-based methods that can be used for few-shot classification, we also report their performance in few-shot experiment settings and compare

| | Methods | PlantVillage | | | | PlantDoc | | | | PlantWild | | | |
|---|---|---|---|---|---|---|---|---|---|---|---|---|---|
| | | Acc | M-P | M-R | M-F1 | Acc | M-P | M-R | M-F1 | Acc | M-P | M-R | M-F1 |
| Fully-supervised | CoOp [40] | 91.00 | 91.09 | 85.94 | 86.89 | 66.73 | 63.94 | 64.85 | 63.92 | 61.16 | 58.66 | 55.95 | 56.26 |
| | CLIP-Adapter [8] | 92.25 | 91.62 | 87.07 | 87.53 | 53.07 | 49.14 | 49.58 | 48.52 | 56.49 | 52.57 | 50.34 | 50.19 |
| | Tip-Adapter-F [38] | 94.05 | 94.30 | 90.57 | 91.94 | 63.56 | 62.69 | 61.34 | 60.91 | 58.83 | 57.66 | 55.03 | 55.80 |
| | T-CNN [34] | 98.80 | **98.65** | **98.69** | **98.67** | 64.36 | 63.54 | 62.03 | 61.09 | 63.61 | 59.29 | 58.84 | 58.70 |
| | DHBP [35] | **98.88** | 98.55 | 98.21 | 98.33 | 65.94 | 65.41 | 63.84 | 62.82 | 65.92 | 60.62 | 60.20 | 59.66 |
| | MVPDR (ours) | 97.72 | 97.40 | 96.44 | 96.83 | **69.90** | **69.92** | **68.97** | **68.87** | **67.20** | **64.03** | **62.64** | **62.84** |
| Few-shot | KNN (N=16) | 62.89 | 65.47 | 65.00 | 59.06 | 41.19 | 44.50 | 43.31 | 41.46 | 36.66 | 36.49 | 35.79 | 33.67 |
| | CoOp* [40] | 72.37 | 70.52 | 70.75 | 67.49 | 57.03 | 56.49 | 56.70 | 55.82 | 51.21 | 47.48 | 50.22 | 47.66 |
| | CLIP-Adapter* [8] | 26.84 | 26.49 | 27.08 | 22.84 | 41.78 | 40.88 | 41.36 | 39.65 | 36.12 | 33.02 | 34.00 | 31.32 |
| | Tip-Adapter-F* [38] | 81.39 | 82.67 | 80.03 | 79.21 | **58.61** | **58.73** | **58.12** | **56.71** | 50.07 | 48.17 | 49.94 | 47.84 |
| | MVPDR* (ours) | **85.20** | **83.45** | **85.23** | **82.89** | 58.42 | 57.84 | 57.81 | 56.50 | **51.80** | **49.01** | **50.27** | **48.21** |

Table 1: Classification results of different methods on plant disease datasets. All the CLIP-based methods take ResNet101 as the backbone of visual encoders. For CoOp, CLIP-Adapter, Tip-Adapter-F and our baseline, we train them with all the training samples in the fully-supervised setting and 16 samples in the few-shot setting.

the results with our baseline. As aforementioned, our versatile baseline not only constructs multiple prototypes for each class in both modalities but also can be applied to various scenarios. Additionally, we evaluate the classification ability of our baseline without training, in comparison to training-free methods such as KNN [20], CALIP [9], CuPL [21] and SuS-X [32].

## 4.2 Main Results

**Comparison with the state-of-the-art methods** Table 1 provides a summary of the key results. Our MVPDR achieves satisfactory performance across the three plant disease datasets. Notably, MVPDR demonstrates a significant advantage on wild image datasets, i.e., PlantWild and PlantDoc. On PlantWild, MVPDR achieves better accuracy than the state-of-the-art T-CNN and DHBP, surpassing them by 3.59% and 1.28% on Accuracy, respectively. The performance gaps become even larger on PlantDoc i.e., 4.54% and 3.96% on Accuracy, respectively. In addition, MVPDR also demonstrates consistent superiority in other metrics, including macro precision, macro recall and macro F1-score.

For the laboratory-based images from PlantVillage, although MVPDR still achieves the highest performance among CLIP-based approaches, it slightly underperforms T-CNN and DBHP. We speculate that the performance gap on PlantVillage is because (i) there is a domain gap between laboratory images and in-the-wild images and (ii) we only fine-tune the prototypes while T-CNN and DBHP train the entire networks. The domain gap is further validated by the result of CLIP (zero-shot) in Table 2. Due to the domain gap, only fine-tuning the feature prototypes in MVPDR is not sufficient. It implies that the features should be adapted to the laboratory images. The unsupervised scenario in Table 2 consolidates that in PlantVillage image features from the same class are very similar, and using KNN can achieve satisfactory performance. This also implies that compared to the laboratory image dataset, in-the-wild datasets, such as PlantDoc and our PlantWild, are more challenging.

**Comparision of training-free classification accuracy** For the unsupervised and zero-shot settings in Table 2, MVPDR only

uses the visual prototypes and textual prototypes to measure similarity, respectively. We evaluate the classification ability of existing training-free baselines, including zero-shot CLIP, Tip-Adapter, CuPL, CALIP, SuS-X and KNN. As indicated by Table 2, KNN surprisingly outperforms the other CLIP-based unsupervised methods across all the datasets. It is worth noting that compared with the few-shot setting in Table 1 KNN performance drops significantly, e.g., almost 15% on Acc in PlantWild. The performance of KNN is still much inferior to the fully supervised results of parametric methods in Table 1.

In the zero-shot setting, there are no training images available for all the methods, and all the competing methods solely rely on the similarity between the visual features of test samples and text descriptions of each class to make predictions. The results in Table 2 (Zero-shot) indicate that MVPDR achieves the best performance between zero-shot methods on the three datasets.

Overall, the experiments in Table 1 and Table 2 indicate that MVPDR performs well when training is permitted or data is not available. In the case that training is not permitted while data is available, KNN is a good option.

## 4.3 Ablation Study

**Weights of loss functions** We investigate various weights $\lambda_1$ and $\lambda_2$ to ensemble the overall loss in Eq. 7. We adjust the two hyper-parameters within the range of 0.0 to 10.0. We first vary $\lambda_1$ while keeping $\lambda_2$ fixed as 1. According to the upper part of Table 3, the classification accuracy reaches its peak when $\lambda_1$ equals 0.1. In the lower part of Table 3, we set $\lambda_1$ to the optimal value 0.1 and then investigate the impact of $\lambda_2$. The results illustrate that better performance is achieved when $\lambda_2$ is set to 0.1. Therefore, the values of $\lambda_1$ and $\lambda_2$ are chosen as 0.1. This indicates that the parameters of visual prototypes require larger gradients than textual prototypes during backpropagation.

**Ratios for logits ensemble** We also conduct ablation studies on the ratios for the logits ensemble in the inference process. We adjust the values of $\alpha$, $\beta$ and $\gamma$ from 0 to 1.0 sequentially. While one ratio is adjusted, the other two are kept fixed. The fixed ratio

| | Methods | PlantVillage | | | | PlantDoc | | | | PlantWild | | | |
|---|---|---|---|---|---|---|---|---|---|---|---|---|---|
| | | Acc | M-P | M-R | M-F1 | Acc | M-P | M-R | M-F1 | Acc | M-P | M-R | M-F1 |
| Unsupervised | KNN (N=1) | 86.92 | 84.29 | 82.71 | 82.87 | 52.47 | 52.52 | 50.84 | 50.80 | 51.07 | 48.55 | 46.95 | 47.18 |
| | KNN (N=5) | 88.84 | 87.20 | 85.04 | 85.33 | **58.61** | **58.70** | **56.62** | **56.58** | 52.65 | 49.51 | 46.72 | 47.01 |
| | KNN (N=10) | **89.40** | **89.15** | **85.56** | **86.33** | 54.06 | 54.35 | 51.78 | 51.42 | **55.18** | **53.91** | **49.29** | **50.01** |
| | Tip-Adapter [38] | 21.28 | 19.30 | 17.99 | 12.61 | 37.23 | 35.82 | 33.99 | 30.90 | 25.02 | 27.95 | 24.02 | 20.12 |
| | MVPDR (ours) | 61.42 | 70.37 | 62.42 | 56.37 | 47.33 | 47.67 | 45.75 | 42.13 | 38.56 | 49.50 | 37.91 | 35.72 |
| Zero-shot | CLIP [22] | 8.42 | 9.42 | 8.26 | 3.68 | 35.84 | 36.89 | 34.66 | 30.95 | 25.94 | 26.40 | 24.98 | 21.41 |
| | CALIP [9] | 8.77 | 8.98 | 8.38 | 3.89 | 36.23 | 35.44 | 33.77 | 30.57 | 26.35 | 26.94 | 24.85 | 21.20 |
| | CuPL [21] | 18.69 | 16.90 | 15.52 | 9.65 | 37.22 | 41.43 | 35.37 | 32.93 | 30.29 | 35.93 | 27.62 | 24.71 |
| | SuS-X [32] | 19.49 | 15.55 | 15.04 | 9.72 | 38.02 | 40.15 | 36.38 | 34.17 | 30.24 | 33.92 | 27.46 | 24.39 |
| | MVPDR† (ours) | **20.19** | **20.23** | **16.85** | **12.55** | **40.00** | **40.38** | **38.37** | **36.45** | **32.77** | **34.92** | **31.85** | **29.04** |

Table 2: Classification results of different training-free methods. For the unsupervised setting, images are available but without labels. For the zero-shot setting, only text descriptions are available.

| $\lambda_1$ | 0.03 | 0.10 | 0.30 | 1.0 | 3.0 | 10.0 |
|---|---|---|---|---|---|---|
| | 65.24 | **66.41** | 65.87 | 65.71 | 65.46 | 65.19 |

| $\lambda_2$ | 0.03 | 0.10 | 0.30 | 1.0 | 3.0 | 10.0 |
|---|---|---|---|---|---|---|
| | 66.85 | **67.20** | 66.55 | 66.41 | 66.11 | 66.09 |

Table 3: Impacts of different weights of the overall loss on PlantWild.

| Backbones | RN50 | RN101 | ViT-B/32 | ViT-B/16 | ViT-L/14 |
|---|---|---|---|---|---|
| PlantVillage | 98.12 | 97.89 | 98.22 | **98.27** | 97.80 |
| PlantDoc | 68.51 | 69.50 | 69.90 | 72.48 | **77.23** |
| PlantWild | 65.30 | 67.20 | 66.85 | 70.76 | **76.18** |

Table 4: Performance of MVPDR with different CLIP backbones.

values are initially set to 0.5 but are later adjusted to the optimal value based on the results. According to Figure 6(a), the best values of $\alpha$, $\beta$, and $\gamma$ are 0.3, 0.5, and 0.5, respectively, as these values lead to the highest accuracy. The results indicate that textual prototypes are slightly more effective in improving performance compared to visual prototypes. In addition, the maximized and averaged logits based on textual prototypes have equal importance in weighting the final probability logits.

**Different cluster numbers**  We investigate the influence of the number of cluster numbers on classification performance. During the construction process of visual prototypes, we utilize the K-Means to cluster image features and employ the cluster centroids as visual prototypes. We respectively choose 1, 2, 4, 8, 16 clusters to train MVPDR. We conduct each experiment 5 times and average the results to obtain the final outcome. The results in Figure 6(b) illustrate that increasing the number of clusters can enhance accuracy. As the number continues to increase, the improvements become less pronounced. This demonstrates that an appropriate number of prototypes can represent diverse features within each category sufficiently.

**Single and multi-modal prototypes**  To offer further insight into prototypes in different modalities, we explore the effectiveness of single-modal prototypes. Instead of fine-tuning multi-modal prototypes simultaneously, we only train prototypes from a single modality (visual or textual). The comparisons are conducted on the three plant disease datasets. As depicted in Figure 6(c), the results of multi-modal prototypes demonstrate superior accuracy compared to those of solely relying on visual or textual prototypes. This implies that prototypes of both modalities contribute to performance. We also notice that textual prototypes exhibit slightly better classification performance than visual prototypes.

**Variants with different backbones**  We conduct experiments with different backbones of CLIP's visual encoder, including variants of ResNet [10] and ViT [6]. Table 4 demonstrates that a larger backbone leads to better performance of MVPDR on PlantDoc and PlantWild. This suggests that larger CLIP backbones are better at identifying plant diseases in complex conditions.

## 4.4 Discussion

**Prototype Visualization**  To better observe the initially constructed prototypes and their changes during the training process, we employ t-SNE [33] to visualize the textual and visual prototypes of MVPDR. The distributions of prototypes before and after training are presented in Figure 7(a) and Figure 7(b) respectively. The numbers from 0 to 4 represent the following plant disease: *apple rust, bean rust, corn rust, ginger leaf spot, peach leaf curl*, and **t** and **v** in legends stand for textual prototypes and visual prototypes, respectively. Figure 7(a) illustrates that visual prototypes from different classes have much closer distances than those of textual prototypes. In addition, the distances between textual and visual prototypes within the same class may not always be smaller than the distances between prototypes belonging to different categories, potentially resulting in misclassification. Figure 7(b) demonstrates that after training, prototypes belonging to different classes are more effectively separated, and the distances between visual and textual prototypes of the same class are significantly minimized.

**Visualization for Explainability**  To explore the explainability of MVPDR, we visualize the attention maps to figure out the focused regions. Given the visual features extracted from input images, we generate their activation maps concerning zero-shot CLIP features, as well as textual and visual prototypes According to the results in Figure 5, we can find the parts of images that have

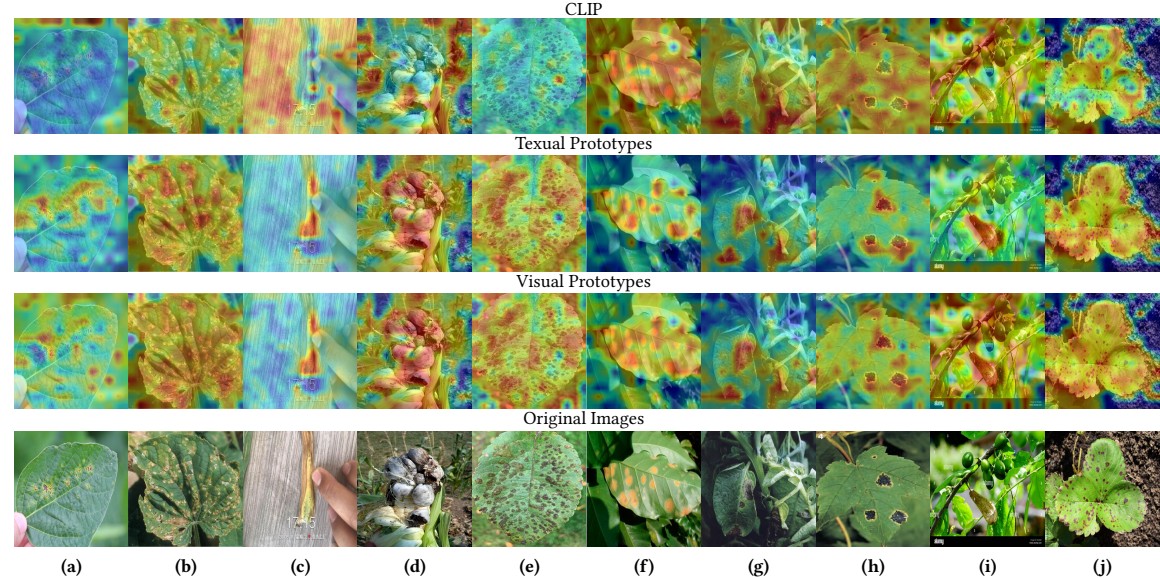

Figure 5: Similarity maps of zero-shot CLIP features and prototypes across 10 plant diseases: (a) bean halo blight; (b) cucumber angular leaf spot; (c) garlic rust; (d) corn smut; (e) apple scab; (f) bean rust; (g) coffee leaf rust; (h) maple tar spot; (i) plum pocket disease; (j) strawberry leaf scorch.

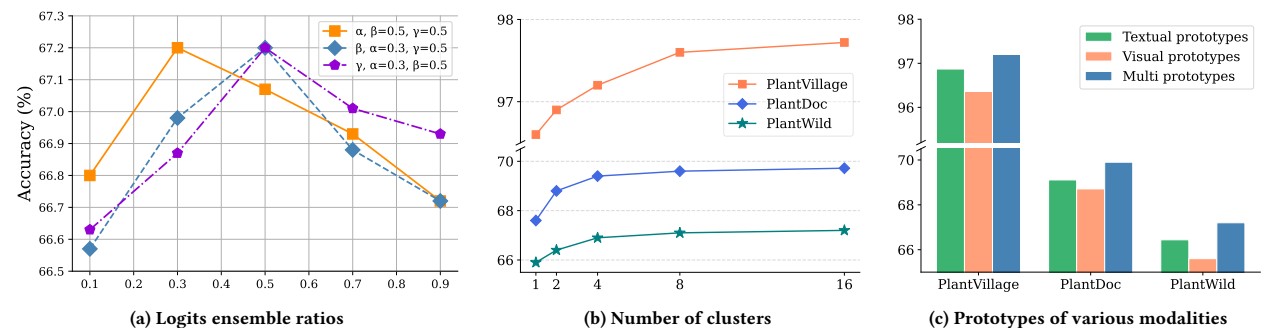

(a) Logits ensemble ratios

(b) Number of clusters

(c) Prototypes of various modalities

Figure 6: Ablation studies on (a) logits ensemble ratios; (b) the number of cluster centroids; (c) prototypes of two modalities.

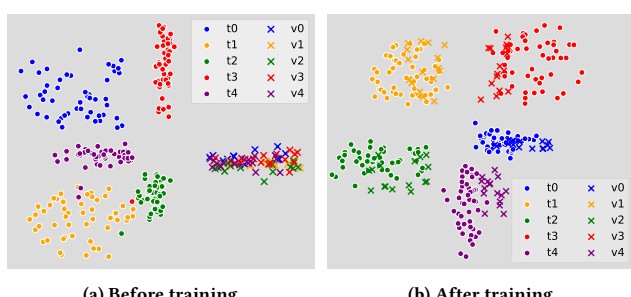

(a) Before training

(b) After training

Figure 7: The t-SNE visualization of prototypes of top-5 performing categories on PlantWild. After training, the visual and textual prototypes of the same category become closer.

high similarities with prototypes and are highly correlated to the predictions. The text features extracted by zero-shot CLIP generally cannot effectively localize the lesion sites of plants, while the trained prototypes show remarkable capability in addressing the infected parts. We observe that the activation maps generated by

textual prototypes are more precise in recognizing the diseased parts compared to visual prototypes, which is also consistent with the quantitative results presented in Figure 6(c).

## 5 CONCLUSION

In this paper, we explore the main challenges of effectively identifying plant disease images with complex backgrounds, including intra-class variance and inter-class discrepancy issues. To investigate these problems, we curate a multimodal in-the-wild plant disease dataset named PlantWild. To the best of our knowledge, PlantWild is currently the largest dataset containing wild plant disease images. In addition, we introduce a versatile multimodal multi-prototype-based plant disease baseline that can be tailed to various testing scenarios, including fully-supervised, few-shot and zero-shot learning. Extensive experiments demonstrate that our baseline outperforms the state-of-the-art on the in-the-wild plant disease data but also the challenges of the newly proposed PlantWild dataset. Moreover, we find that our baseline MVPDR can effectively localize the positions of plant disease lesions, showcasing its potential for disease detection tasks.

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
