# OpenReview forum: "Benchmarking In-the-wild Multimodal Disease Recognition and A Versatile Baseline"
_acmmm.org/ACMMM/2024/Conference — MM2024 Poster_

### Official Review · Reviewer_6E1H · 2024-05-03

**Rating:** 3
**Confidence:** 3

**Summary:**

The main contribution of this paper is to propose PlantWild dataset, which is currently the largest dataset containing wild plant disease images, and benchmark state-of-the-art methods in the conventional classification task as well as few-shot and training-free situations.

**Strengths:**

1. The paper proposes a new wild plant dataset, which is meaningful.
2. The writting is fine.
3. Various testing scenarios, including fully-supervised, few-shot and zero-shot learning, are constructed for PlantWild. It is meaningful for different applicaition situations.

**Limitations:**

1. I wonder how to determine that for the new dataset, there is overlap with PlantDoc dataset and PlantVillage or not? If there is overlap, what is the ratio?
2.   The work is meaningful , while the innovation can be improved.

**Suitability:**

2

---

### Official Review · Reviewer_a96K · 2024-05-22

**Rating:** 5
**Confidence:** 3

**Summary:**

This paper proposed a new in-the-wild or real world multimodal plant disease dataset. They use ChatGPT to generate text description for each of the classes in the dataset. Besides, they also proposed a multimodal baseline, named MVPDR to address the small inter-class discrepancy and large intra-class variance issues. The proposed model outperforms SOTA models in term of accuracy and visualization performance.

**Strengths:**

This paper compiles a comprehensive dataset from various online platforms to create one of the largest real-world plant disease datasets. In addition, the authors present a new methodology that uses visual and textual prototypes as classifiers to improve the accuracy of real-world plant disease identification tasks. The paper includes sufficient, though not conclusive, ablation studies to validate the proposed baselines.

**Limitations:**

1. The dataset compiled in this paper includes images sourced from online search engines such as Google Images and Baidu. Since these images can be uploaded by any individual, their validity and reliability are potentially questionable. The study addresses this concern by implementing data cleaning and validation processes conducted by five annotators and an expert. However, the detailed procedure of this validation process is not fully documented in the paper. For enhanced credibility, it would be advantageous to have the dataset validated by experts in plant pathology. Additionally, the authors could provide a breakdown of the composition of images from each source. Although this dataset is currently the largest in the plant disease domain, it does not include other available annotated datasets, such as those from Kaggle.

2. In the fully supervised setting presented in Table 1, the proposed model exhibits slightly lower performance compared to state-of-the-art (SOTA) models. The authors suggest that this may be due to the use of frozen CLIP backbones. It is understandable that there is a domain gap between laboratory and real-world images. However, it is somewhat unexpected that the performance on laboratory images is lower than on real-world images, as laboratory images are typically considered easier samples. The original weights of the CLIP model are trained on a large dataset for general object identification. To further validate the performance of their proposed model, the authors could consider fully training their model end-to-end (unfreeze CLIP backbones).

3. For the ablation studies for the weight of loss function, the author set weight for both text losses but not visual loss (eq 4). The authors could also set weight for the visual loss to fully analyse the performance of the overal loss (eq 7)

4. The authors extract their own textual descriptions based on information from Wikipedia and ChatGPT. However, there are no experimental results demonstrating that these extracted textual descriptions are superior to the original disease labels. To address this, the authors could conduct an ablation study to evaluate the performance of the extracted textual descriptions in comparison to the original text labels.

**Suitability:**

3

---

### Official Review · Reviewer_bnG8 · 2024-05-24

**Rating:** 4
**Confidence:** 3

**Summary:**

Plants frequently encounter various threats from diseases caused by bacteria, pests, and viruses. According to the Food and Agriculture Organization of the United Nations, these plant diseases lead to annual losses of around 220 billion dollars. Identifying plant diseases accurately is crucial for reducing damage and preventing their spread. This paper proposes a large-scale multimodal in-the-wild plant disease recognition dataset which can be regarded as an ideal testbed for evaluating disease recognition methods in the world. The proposed dataset contains not only diseased and healthy plant images but also multiple text descriptions for each class. The image data are crowdsourced from diverse internet sources and the descriptions of each class are obtained from Wikipedia and GPT-3.5. In addition, this paper also propose a versatile multimodal multi-protoypebased plant disease baseline model to not only classify disease but also recognize diseases in few-shot or training-free scenarios. This paper conducted extensive experiments. All results demonstrate that the proposed baseline outperforms the state-of-the-art on the in-the-wild plant disease dataset but also the challenges of the newly proposed PlantWild dataset. Moreover, the baseline MVPDR can effectively localize the positions of plant disease lesions, showcasing its potential for disease detection tasks. This work may contribute to the study of automatic plant disease recognition.

**Strengths:**

This paper proposes an in-the-wild multimodal plant disease recognition dataset that advances the field of Object recognition. Extensive benchmarking results validated that the proposed in-the-wild multimodal dataset sets many new challenges to the plant disease recognition task. Additionally, the paper makes valuable theoretical contributions and provides detailed technical explanations, ensuring transparency and reproducibility. The motivation of this study is strong, and this paper is well-organized to clearly present its novelty and contribution.

**Limitations:**

(1) The proposed Plantwild dataset has a large difference across different classes in the number of samples (form 44 to 589 for each class) which may cause data imbalance.
(2) This paper uses grouping technique K-means to construct visual prototype which makes each class have K class centers. However, enabling all the classes have the same number of class centers is obviously unreasonable. In additional, some classes have very few samples which make it unreliable to obtain K class centers.
(3) In lines 325 to 327, the author mentioned that 'If the two annotators cannot reach a consensus on a particular image, we invite another expert to review and correct the annotations.' It is possible that two annotators will agree on an incorrect annotation for a particular image. Therefore, the proposed PlantWild Dataset lacks authority.

**Suitability:**

2

---

### Official Review · Reviewer_jeYE · 2024-05-24

**Rating:** 4
**Confidence:** 3

**Summary:**

This paper focuses on the problem of plant disease classification using in-the-wild images. To overcome the problem of small inter-class discrepancy and large intra-class variance, the authors proposes an in-the-wild multimodal plant disease recognition dataset. The authors claim that this is currently the largest in-the-wild plant disease dataset in terms of image numbers and disease classes. They also propose a baseline method, specifically using the CLIP encoder to extract features from images and texts of each category, and then constructing visual and textual prototypes. Experimental results show that the baseline outperforms the state-of-the art on the in-the-wild plant disease data.

**Strengths:**

1. This paper focuses on the problem of plant disease classification using in-the-wild images, which is a challenging but important problem. Besides, since the dataset is constructed in two modalities, the topic of this paper exactly coincides with this conference.
2. The proposed dataset is an important contribution to the field of agriculture. Besides, considering that the images in this dataset have small inter-class discrepancy and large intra-class variance, the dataset will promote technological advancement in computer vision.
3. The prototype based classification method proposed in this paper achieves satisfactory results, which provides a good baseline for this problem.
4. The paper is written clearly and fluently, with very detailed explanations of the principles.

**Limitations:**

1. The authors have proposed a more comprehensive and detailed dataset, but there lack comparisons with other similar datasets in the same field about annotation types, sample sizes, etc. Besides, I suggest the authors also explain the advantages/disadvantages of the new dataset compared with previous ones.
2. The prototypes proposed by the authors do not fully utilize the mentioned descriptions, as shown in the detailed information provided in Figure 2.
3. I suggest the authors create a table to visually present the innovative aspects of the proposed new dataset.

**Suitability:**

3

---

### Meta-Review · Area_Chair_cxDA · 2024-07-01

**Recommendation:** Accept (Poster)
**Confidence:** 4

**Metareview:**

All reviewers have agreed on the acceptance of the paper. The merits include dataset contribution, clear presentation and good baseline method as a potential solution.